# Polyp Segmentation Using Wavelet-Based Cross-Band Integration for Enhanced Boundary Representation

## Abstract

Accurate polyp segmentation is essential for early colorectal cancer detection, yet achieving reliable boundary localization remains challenging due to low mucosal contrast, uneven illumination, and color similarity between polyps and surrounding tissue. Conventional methods relying solely on RGB information often struggle to delineate precise boundaries due to weak contrast and ambiguous structures between polyps and surrounding mucosa. To establish a quantitative foundation for this limitation, We analyzed polyp–background contrast in the wavelet domain, revealing that grayscale representations consistently preserve higher boundary contrast than RGB images across all frequency bands. This finding suggests that boundary cues are more distinctly represented in the grayscale domain than in the color domain. Motivated by this finding, we propose a segmentation framework that integrates grayscale and RGB representations through complementary frequency-consistent interaction, enhancing boundary precision while preserving structural coherence. Extensive experiments on four benchmark datasets demonstrate that the proposed approach achieves superior boundary precision and robustness compared to conventional methods.

## 1 Introduction

Accurate polyp segmentation is vital for early colorectal cancer detection but remains difficult due to low mucosal contrast, uneven illumination, and strong visual similarity between polyps and surrounding mucosa [1]. These conditions blur polyp boundaries, particularly in small or flat cases, making reliable delineation challenging. Although recent boundary-aware methods improve edge perception, their dependence on RGB inputs limits robustness under contrast variations. To investigate this limitation, we performed a wavelet-based contrast analysis between RGB and grayscale images. The contrast index (CI), defined as $CI = |\mu_{\text{polyp}} - \mu_{\text{background}}|/(\mu_{\text{polyp}} + \mu_{\text{background}} + \epsilon)$, measures the distinction between polyp and background regions using the mean of absolute wavelet coefficients. As shown in Fig. 1, grayscale consistently achieves higher CI across all sub-bands, indicating that boundary cues are more distinct in the intensity domain. Building on this evidence, we propose a segmentation approach that integrates grayscale and RGB representations through frequency-band interaction. By encouraging interaction between corresponding wavelet sub-bands of both modalities, contrast-rich grayscale features refine RGB-derived spatial structures, improving boundary precision while preserving overall coherence.

## 2 Related Work

Polyp segmentation has become a key task in computer-aided colorectal cancer screening, where precise boundary delineation is essential for accurate diagnosis and treatment planning. A variety of

Submitted to 39th Conference on Neural Information Processing Systems (NeurIPS 2025). Do not distribute.

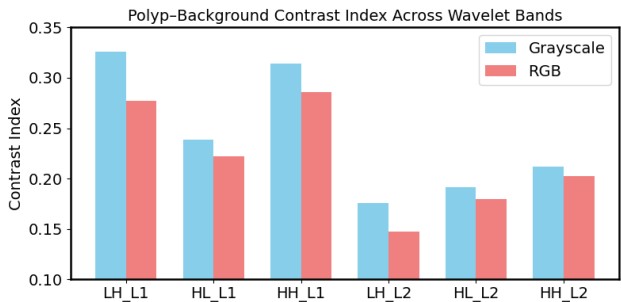

Figure 1: Structural contrast comparison between RGB and grayscale images in the wavelet domain, showing consistently higher contrast for grayscale across all detail sub-bands.

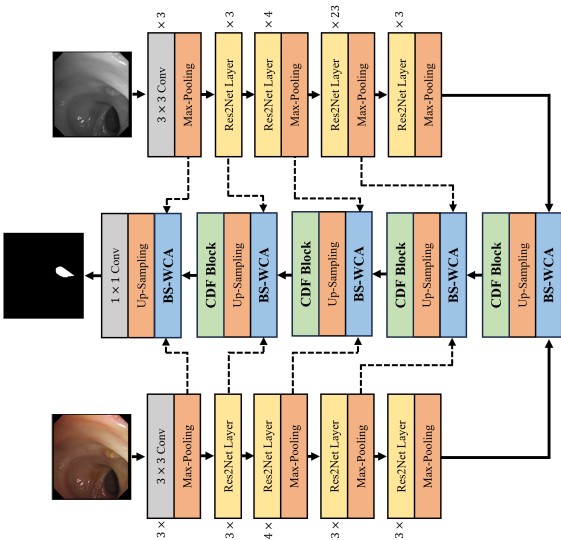

Figure 2: Proposed wavelet-based cross-band integration framework that fuses frequency-consistent information from RGB and grayscale features for enhanced boundary representation.

deep learning–based models have been developed for polyp segmentation [11, 15, 14, 8, 5]. Among these, boundary-aware models such as PraNet [6], CaraNet [10], MEGANet [4], and Polyper [12] aim to improve boundary localization through attention- and edge-guided mechanisms for more accurate polyp delineation. Nevertheless, most of these approaches primarily rely on RGB representations, which capture chromatic appearance but insufficiently describe structural contrast.

## 3 Method

The proposed model adopts a dual-encoder structure designed to leverage the complementary characteristics of RGB and grayscale modalities. Each encoder is based on Res2Net [7] and extracts hierarchical feature representations: the RGB encoder captures chromatic and textural cues, while the grayscale encoder focuses on contrast-driven structural patterns that are effective for boundary discrimination. Features from corresponding encoder stages are processed within the decoder through two key components: the Band-Specific Window Cross-Attention (BS-WCA) module and the Cascade Dilated Fusion (CDF) block. The BS-WCA module performs frequency-aligned interaction between the RGB and grayscale features by selectively exchanging information across identical wavelet sub-bands, allowing high-frequency grayscale details to refine RGB-derived structural features. The CDF block then integrates the refined multi-scale features through dilated convolutions, preserving both fine-grained boundary precision and global contextual consistency. As illustrated

Table 1: Quantitative comparison of polyp segmentation methods across four benchmark datasets, evaluated by mean Dice and IoU scores (mean ± standard deviation) averaged over 10 random runs.

| Methods | Kvasir | | ClinicDB | | ColonDB | | ETIS | |
|---|---|---|---|---|---|---|---|---|
| | mDice | mIoU | mDice | mIoU | mDice | mIoU | mDice | mIoU |
| **Ours** | **0.885 ± 0.021** | **0.822 ± 0.019** | **0.926 ± 0.014** | **0.862 ± 0.023** | **0.913 ± 0.021** | **0.840 ± 0.042** | **0.922 ± 0.029** | **0.821 ± 0.029** |
| Polyper | 0.867 ± 0.014 | 0.796 ± 0.020 | 0.914 ± 0.019 | 0.841 ± 0.021 | 0.868 ± 0.042 | 0.796 ± 0.035 | 0.888 ± 0.047 | 0.760 ± 0.047 |
| MEGANet | 0.863 ± 0.011 | 0.802 ± 0.018 | 0.909 ± 0.011 | 0.801 ± 0.077 | 0.802 ± 0.083 | 0.704 ± 0.059 | 0.747 ± 0.097 | 0.598 ± 0.077 |
| CRCNet | 0.879 ± 0.015 | 0.815 ± 0.018 | 0.910 ± 0.052 | 0.854 ± 0.018 | 0.866 ± 0.064 | 0.800 ± 0.046 | 0.665 ± 0.027 | 0.582 ± 0.129 |
| CaraNet | 0.727 ± 0.023 | 0.628 ± 0.035 | 0.836 ± 0.006 | 0.702 ± 0.014 | 0.760 ± 0.056 | 0.651 ± 0.038 | 0.784 ± 0.078 | 0.633 ± 0.105 |
| ConvSegNet | 0.856 ± 0.008 | 0.765 ± 0.022 | 0.902 ± 0.020 | 0.810 ± 0.025 | 0.884 ± 0.033 | 0.756 ± 0.044 | 0.859 ± 0.055 | 0.667 ± 0.077 |
| DUCKNet | 0.818 ± 0.016 | 0.751 ± 0.019 | 0.878 ± 0.026 | 0.791 ± 0.033 | 0.683 ± 0.161 | 0.570 ± 0.073 | 0.383 ± 0.205 | 0.321 ± 0.099 |
| PraNet | 0.650 ± 0.021 | 0.524 ± 0.032 | 0.793 ± 0.032 | 0.648 ± 0.030 | 0.784 ± 0.063 | 0.620 ± 0.037 | 0.666 ± 0.090 | 0.423 ± 0.092 |
| UNet | 0.775 ± 0.013 | 0.668 ± 0.025 | 0.855 ± 0.019 | 0.762 ± 0.024 | 0.802 ± 0.083 | 0.704 ± 0.059 | 0.549 ± 0.186 | 0.382 ± 0.099 |

in Fig. 2, the integration of BS-WCA and CDF within a dual-encoder–decoder framework enables model to exploit contrast-rich intensity information while maintaining the contextual advantages of RGB representations, resulting in more stable and accurate polyp boundary segmentation.

## 4 Experimental Results

**Datasets and Evaluation** The experiments utilized four widely used polyp segmentation datasets: Kvasir-SEG (1,000 images) [9], CVC-ClinicDB (612 images) [3], CVC-ColonDB (380 images) [2], and ETIS (196 images) [13]. All datasets were partitioned into training, validation, and testing sets to ensure consistent evaluation across varying data distributions. Performance was measured using the Dice coefficient and Intersection over Union (IoU), following standard practices in polyp segmentation.

**Implementation Details** All experiments were implemented using PyTorch and Torchvision and conducted on a single NVIDIA GPU (CUDA 11.7, Ubuntu 20.04, Python 3.9, PyTorch 1.13, Torchvision 0.14). The proposed model and all baseline models were trained and evaluated with a batch size of 8.

**Results.** As shown in Table 1, the proposed method consistently outperforms existing approaches across all four benchmark datasets in terms of both Dice and IoU scores. The improvement is particularly evident in the overall average performance, demonstrating the benefit of jointly leveraging grayscale and RGB representations. Compared to recent conventional models, the proposed model achieves higher mean Dice and IoU values while maintaining stable performance across datasets of varying scales and imaging conditions. These results indicate that the integration of intensity-based and color-based representations provides a more comprehensive feature space for polyp segmentation.

## 5 Discussion

This study primarily evaluates region-level segmentation performance using Dice and IoU. To better understand the behavior of the proposed model, further analyses should include boundary-sensitive metrics for contour accuracy, qualitative comparisons across imaging conditions, and frequency-domain ablations to assess modality contributions. Such evaluations would provide a more comprehensive view of how grayscale–RGB integration influences segmentation performance.

## 6 Conclusion

This work introduced a dual-encoder segmentation framework that integrates grayscale and RGB representations through frequency-band interaction. Although the proposed model does not explicitly model boundaries, its design naturally facilitates boundary-aware representation through high-frequency information exchange. The design effectively combines contrast-rich intensity cues with color-based features, achieving consistent performance improvements across four benchmark datasets. Future work will extend this framework with boundary-sensitive evaluations and frequency-domain analyses to further clarify its contribution to robust polyp boundary segmentation.

# 7 Potential Negative Societal Impact

While the proposed method aims to improve segmentation reliability in medical imaging, several potential risks should be noted. Generalization across imaging domains may be limited, as variations in endoscopic devices or acquisition settings can influence performance consistency. In addition, limited diversity in publicly available training data could result in biased or uneven segmentation outcomes across populations. To mitigate these risks, careful cross-domain validation, transparent dataset reporting, and responsible integration into medical workflows are essential to ensure equitable and trustworthy application.

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
