# OpenReview forum: "Polyp Segmentation Using Wavelet-Based Cross-Band Integration for Enhanced Boundary Representation"
_EurIPS.cc/2025/Workshop/MedEurIPS — EurIPS 2025 Workshop MedEurIPS Submission_

### Official Review · Reviewer_TpSW · 2025-10-25
**Polyp Segmentation Using Wavelet-Based Cross-Band Integration for Enhanced Boundary Representation**

**Rating:** 6
**Confidence:** 3

**Review:**

The paper introduced a polyp segmentation method that considered both RGB and grayscale information, as well as their interaction. This idea was based on a preliminary experiment that showed the potential of grayscale images for enhancing boundary delineation. Although the motivation was interesting and the experiments were extensive, the method could be described more clearly.

Pros:
- The motivation and the preliminary experiment were interesting.
- The experimental results were extensive, with several baseline methods.

Cons:
- Some components of the method were not explained clearly, like BS-WCA and CDF.
- The metrics were all pixel-based. Some metrics, like HD95, could show more insights.

---

### Official Review · Reviewer_3kJ7 · 2025-10-29
**Logically structured paper with original discovery and method for polyp segmentation, but results are unconvincing because the nnUNet baseline, scored in the litterature on the same public datasets, is missing here for comparison.**

**Rating:** 5
**Confidence:** 5

**Review:**

Logically structured paper with original discovery and method for polyp segmentation, but results and importance are unconvincing because the nnUNet baseline, avalaible in the litterature for the same public datasets, is missing here for comparison.

In this paper (https://arxiv.org/abs/2407.04353) the U-net and nnUnet baseline are given for the CVC-ClinicDB
dataset. The DSC goes from 0.813 (U-Net) to 0.941 (nnUnet). On the same dataset the authors report 0.762 DSC
for U-Net and 0.926 for their method. I'm therefore not convinced about the improvement until I see a direct comparison
with nnU-Net. And if improvement there is, the authors have to justify why it is clinically meaningful.

Pro: simplicity and originality of the idea (combine grayscale and RGB images
       Good motivation provided with a discovery of the stronger contrast in grayscale image.
       Nice flow of the paper

Cons: Results are unconvincing as the nnUnet baseline is missing ('UNet' alone is not enought)
          The first sentence of abstract, introduction and related work express the same idea.
          How where datasets divided into training/val/test? What is meant by "random runs"'?

---

### Decision · Program_Chairs · 2025-10-31

**Decision:**

Accept (Poster)

**Comment:**

Both reviewers find the paper well motivated and logically structured, with an interesting observation that grayscale representations enhance boundary contrast for polyp segmentation.